

# Using natural history collections to investigate changes in pangolin (Pholidota: Manidae) geographic ranges through time

Emily Buckingham[1,2,*], Jake Curry[2,*], Charles Emogor[3,4,5], Louise Tomsett[1] and Natalie Cooper[1]

[1] Department of Life Sciences, Natural History Museum, London, London, United Kingdom
[2] Department of Life Sciences (Silwood Park), Imperial College London, Ascot, United Kingdom
[3] Department of Zoology, University of Cambridge, Cambridge, United Kingdom
[4] IUCN SSC Pangolin Specialist Group, Zoological Society of London, London, UK
[5] Wildlife Conservation Society, Nigeria Program, Calabar, Nigeria
[*] These authors contributed equally to this work.

Corresponding author
Natalie Cooper, natalie.cooper@nhm.ac.uk

## ABSTRACT

Pangolins, often considered the world's most trafficked wild mammals, have continued to experience rapid declines across Asia and Africa. All eight species are classed as either Vulnerable, Endangered or Critically Endangered by the International Union for Conservation of Nature (IUCN) Red List. Alongside habitat loss, they are threatened mainly by poaching and/or legal hunting to meet the growing consumer demand for their meat and keratinous scales. Species threat assessments heavily rely on changes in species distributions which are usually expensive and difficult to monitor, especially for rare and cryptic species like pangolins. Furthermore, recent assessments of the threats to pangolins focus on characterising their trade using seizure data which provide limited insights into the true extent of global pangolin declines. As the consequences of habitat modifications and poaching/hunting on species continues to become apparent, it is crucial that we frequently update our understanding of how species distributions change through time to allow effective identification of geographic regions that are in need of urgent conservation actions. Here we show how georeferencing pangolin specimens from natural history collections can reveal how their distributions are changing over time, by comparing overlap between specimen localities and current area of habitat maps derived from IUCN range maps. We found significant correlations in percentage area overlap between species, continent, IUCN Red List status and collection year, but not ecology (terrestrial or arboreal/semi-arboreal). Human population density (widely considered to be an indication of trafficking pressure) and changes in primary forest cover, were weakly correlated with percentage overlap. Our results do not suggest a single mechanism for differences among historical distributions and present-day ranges, but rather show that multiple explanatory factors must be considered when researching pangolin population declines as variations among species influence range fluctuations. We also demonstrate how natural history collections can provide temporal information on distributions and discuss the limitations of collecting and using historical data.

## INTRODUCTION

Pangolins (Pholidota: Manidae) are insectivorous mammals found in parts of Africa and Asia (*Hua et al., 2015*). They are considered the world's most trafficked wild mammal due to significant consumer demand for their scales and meat (*Challender, Harrop & MacMillan, 2015*; *Cheng, Xing & Bonebrake, 2017*). Historically, both African and Asian species have locally been traded for consumption, but as local population levels have declined in parts of Asia (*Irshad et al., 2015*; *Challender, Nash & Waterman, 2020*; *Wu et al., 2004*), researchers have documented a shift in demand from Asia for African pangolins (*Challender, Harrop & MacMillan, 2015*; *Heinrich et al., 2016*) which is believed to be the leading cause of declines in African pangolin populations (*IUCN, 2020*). In addition, habitat destruction and slow reproductive rates restrict the rate at which pangolins can recover from overexploitation (*Heinrich et al., 2016*), and issues with disease control and dietary husbandry limit the success of captive breeding programs (*Hua et al., 2015*). As all eight species are listed as either Vulnerable, Endangered or Critically Endangered by the International Union for Conservation of Nature (IUCN; *IUCN, 2020*; *Heinrich et al., 2016*; Cheng et al., 2017), a better understanding of the threats to, and conservation status of, pangolins is therefore paramount for protecting them.

Despite listing all eight pangolin species under Appendix I of the Convention on International Trade in Endangered Species (CITES) since 2016, pangolin trafficking has often been poorly documented and not effectively monitored, if detected at all (*Heinrich et al., 2016*), so the actual impact of the global illegal trade on pangolin populations and distributions remain unknown. Furthermore, the lack of adequate modern-day records of pangolin presence makes it hard to investigate geographic changes and consequently predict their extinction risks. Effective species threat assessment relies heavily on changes in the geographical distribution of the species over time (criterion B, IUCN Red List Categories and Criteria; *IUCN, 2020*). Thus, understanding how pangolin distributions have changed in the past decades will provide more insights into their possible population declines and ultimately inform science-based conservation actions.

One possible solution to better understand the conservation status of pangolins is to compare their past and current distributions to highlight regions that may have previously been targeted by traffickers, i.e., regions where species ranges have become smaller, without any obvious associated anthropogenic changes. Museum specimen records can provide both the temporal and spatial data needed to analyse distributional trends (*Boakes et al., 2010*; *Pyke & Ehrlich, 2010*; *Lister et al., 2011*; *McLean et al., 2016*; *Meineke et al., 2018*), without relying on expensive, time consuming, long-term surveys (*Newbold, 2010*; although museum records have other limitations which we highlight in the Discussion). As a result, historical specimen records can be readily used to improve current threat evaluations for pangolins given the paucity of modern data.

Using pangolin museum specimen records from the Global Biodiversity Information Facility (GBIF; *GBIF, 2019* and the Natural History Museum, London (NHM), with

**Table 1 Details of pangolin dataset.** The table shows the number of specimens used in the analyses where certainty scores ≥50% and extent <50 km. Numbers in brackets reflect unique collection localities.

| Species | Continent | IUCN Red List status | Ecology | Number of specimens | Number of specimens with years |
|---|---|---|---|---|---|
| *Manis crassicaudata* | Asia | Endangered | Terrestrial | 12 (11) | 8 (8) |
| *Manis culionensis* | Asia | Critically Endangered | Semi-arboreal | 9 (6) | 3 (2) |
| *Manis javanica* | Asia | Critically Endangered | Semi-arboreal | 48 (37) | 30 (28) |
| *Manis pentadactyla* | Asia | Critically Endangered | Terrestrial | 68 (30) | 21 (18) |
| *Phataginus tetradactyla* | Africa | Vulnerable | Arboreal | 17 (13) | 8 (8) |
| *Phataginus tricuspis* | Africa | Endangered | Semi-arboreal | 97 (49) | 25 (22) |
| *Smutsia temminckii* | Africa | Vulnerable | Terrestrial | 11 (9) | 6 (6) |
| *Smutsia gigantea* | Africa | Endangered | Terrestrial | 7 (7) | 5 (5) |

geographic range maps and habitat classifications by the IUCN SSC Pangolin Specialist Group (*IUCN, 2020*), we produced area of habitat (AOH) maps representing present-day ranges of pangolins and then investigated geographic range contractions in pangolins over the last 150 years by examining overlaps between historical specimen localities and the AOH present-day ranges. We also investigated the effects of land-use change as a proxy for habitat loss, and human population size changes as a proxy for increased exploitation (*Woodroffe, 2000*).

## MATERIALS AND METHODS

### Data collection

We downloaded locality data for Manidae specimens from GBIF (GBIF 2019; https://doi.org/10.15468/dl.o1t25o), selecting only preserved specimens with taxonomy and locality information without flagged issues. These data came from 42 international museums, excluding the Natural History Museum, London (NHM). We excluded GBIF records from the NHM, and instead added more complete records from NHM, including data that were not previously available on GBIF. We also excluded specimens with uncertain taxonomy, or genus-level records only, and corrected the taxonomy using Mammal Species of the World (*Wilson & Reeder, 2005*). We downloaded geographic range map polygons (created in 2019), habitat classifications and IUCN Red List statuses for each pangolin species from the IUCN Red List website (*IUCN, 2020*). IUCN geographic range maps represent the best available depiction of the historical, present and projected distribution of a taxon, as judged by expert assessors based on their knowledge of the taxon and the available data. Species are classed as extant in an area if the species is known or thought very likely to currently occur in the area, which encompasses localities with current or recent (last 20–30 years) records where suitable habitat at appropriate altitudes remains (*IUCN, 2020*). We also used *Challender, Nash & Waterman (2020)* to classify each species as primarily terrestrial or arboreal/semi-arboreal based on whether they are ground or tree-dwelling respectively (Table 1).

### Georeferencing

For the NHM locality data, all georeferencing was carried out by one of us (EB) using the GBIF Best Practice guidelines (*Chapman & Wieczorek, 2006*). Each location was found in Google Maps, and decimal latitude and longitude coordinates for the midpoint of the locality were recorded. We also recorded an extent (in m) from the midpoint to the furthest place that could still be considered to be within the locality as a measure of error (i.e., the point-radius method; *Wieczorek, Guo & Hijmans, 2004*), along with a certainty score (0, 25, 50, 75 or 100%) describing how confident we were that the locality truly fell within the recorded extent. For the GBIF data we followed the same protocol, except where localities had already been georeferenced we used the existing coordinates and just estimated the extent (in m).

Before analyses we checked all records for data quality. We excluded specimens with extents >1,000 km and specimens from zoos or where we suspected the locality data were the shipping, rather than collection, location. Specimens with localities more than 100 km outside their current geographic range were also checked. Where this was a clear-cut taxonomic error we corrected the taxonomy, for example any *Manis* species found in Sri Lanka must be *Manis crassicaudata* as no other members of the genus are found there. If there was no obvious taxonomic error we assumed this was a recording error and omitted the specimen from further analyses. Details of excluded and modified specimens are shown in Tables S1 and S2.

To incorporate georeferencing error, for each specimen locality we created a point-radius polygon from the error extent for that locality, using the R packages sf and sfe (*Pebesma, 2018*; *Curry, 2019*; Fig. 1). The point-radius polygon is a circle around the locality with a radius equal to the error extent of the locality.

### Area of habitat (AOH) maps

We used the known habitat preferences, altitudinal limits and 2019 geographic range polygons all sourced from the IUCN Red List website (*IUCN, 2020*) to create AOH maps for each pangolin species. Trends in species AOH over time can be useful in obtaining estimates of their population decline under the IUCN Red List criteria and in assessing levels of fragmentations of pangolin habitats (*IUCN, 2020*; *Brooks et al., 2019*). No elevation data could be found for *Phataginus tricuspis* and *P. tetradactyla* so just the habitat preferences were used to create the AOH maps for both of these species. Minimum altitudinal limits were not present for either *Smutsia* species, so the value was assumed to be 0 (as is the case with all other species where this data was available; *Brooks et al., 2019*). We downloaded a 2018 land cover classification raster (i.e., gridded data) at 100m horizontal resolution from the Copernicus Climate Change Service version 2.1.1 (*Buchhorn et al., 2019*) and an elevation raster from WorldClim (*Fick & Hijmans, 2017*) at 1 km resolution. Following the methods outlined by *Brooks et al. (2019)*, we used the raster package in R (*Hijmans, 2018*) to crop both the elevation and land-use rasters for each species using the IUCN range maps to define the extents. We then used the raster calculator in QGIS version 3.14 (*QGIS.org, 2020*) to create binary rasters using the known habitat preferences and altitude limits of

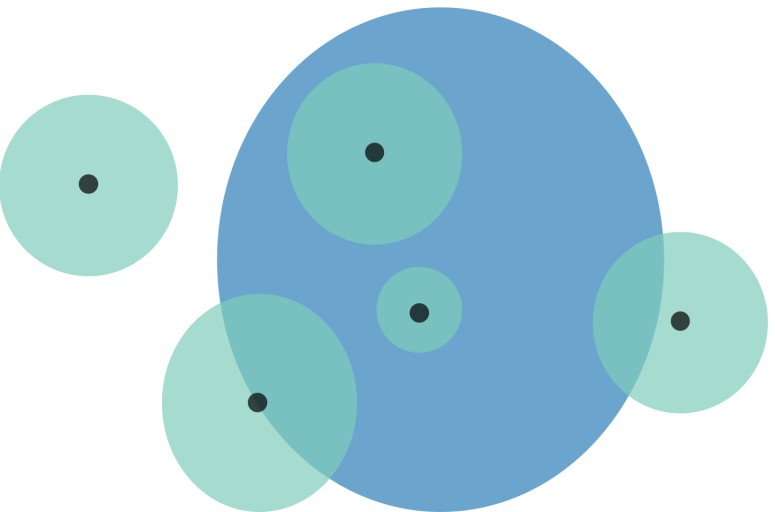

**Figure 1 Schematic representation of overlaps for a hypothetical species.** Black points are the localities that specimens were collected from. These locality points are surrounded by a point-radius polygon (green) to represent possible error in georeferencing, the radius of which is equivalent to the error extent recorded during georeferencing. The area of habitat (AOH) polygon for the species is in blue. Overlaps for this scenario would be as follows: (1) four of the five specimen localities (including error extents) fall within the AOH, so percentage locality overlap is 80%; and (2) the percentage area overlap between the point-radius polygons and AOH polygon is (from left to right) 0%, 50%, 100%, 100%, and 25%, meaning that mean percentage area overlap is 55%.

each species, with the following formula (where XX is a species):

$$(``XX\_elevation" < 0) * 0 + ((``XX\_elevation" >= 0)$$
$$AND(``XX\_elevation" <= 2015)) * 1 + (``XX\_elevation" > 2015) * 0. \tag{1}$$

A value of 1 in the binary raster is therefore suitable habitat and 0 is unsuitable. These binary elevation and land-use rasters were then resampled and multiplied together for each species in R to create the binary suitable/unsuitable AOH maps.

### Anthropogenic data

Geographical analyses are often based on summarising data within grid cells of a defined size, then using values for each grid cell as input data for downstream analysis (e.g., *Derrick, Cheok & Dulvy, 2020*). We downloaded historical and current human population size data as population count and population density from the History Database of the Global Environment (HYDE v3.2.1; *Goldewijk et al., 2017*) at 5′ or 0.083 degree resolution. For land-use change, we downloaded historical land-use states data at 0.25 × 0.25 degree spatial resolution from the Land-Use Harmonization (LUH2) project (LUH2 v2 h; http://luh.umd.edu/data.shtml; *Hurtt et al., 2020*). These data represent the fraction of each 0.25 × 0.25 degree grid cell covered by each land-use type. The land-use types we focussed on were forested primary land (primf; all forested areas including scrublands and savanna, previously undisturbed by any human activities post 850), non-forested primary land (primn; all non-forested areas including grasslands and wetlands, previously

undisturbed by any human activities post 850) and urban land (urban). Both HYDE and LUH2 data were based on the high data-driven land-use reconstructions from HYDE (*Goldewijk et al., 2017*).

For the human population size and the land-use data, we calculated the change in population size or land-use cover (but not both) for each $0.25 \times 0.25$ degree grid cell between 2015 (present-day) and 1850, 1900 or 1950 in turn. We used 2015 as our present-day baseline as these were the most recent data available (*Goldewijk et al., 2017*; *Hurtt et al., 2020*), and used 1850 as our earliest historical date as most specimens were collected between 1850 and 1950 (ideally we would extract values for the year each specimen was collected, but we do not have year data for all specimens). For each specimen and time period, we then extracted the mean change in every population size or land-use variable across the $0.25 \times 0.25$ degree grid cells occupied by the specimen's point radius polygon. We used these values in our analyses below. Note that specimen MVZ_MVZ:Mamm:125554 was excluded from these analyses because it comes from an extremely small island on the P'eng-hu peninsula and there were not sufficient land-use or human population data within this locality for overlaps to be calculated.

All data required to repeat the analyses are available on the NHM Data Portal (data.nhm.ac.uk; *Buckingham et al., 2019*).

## Analyses

We used R version 3.6.2 (*R Core Team, 2019*) for all analyses, and the R packages sf (*Pebesma, 2018*), raster (*Hijmans, 2018*), rgdal (*Bivand, Keitt & Rowlingson, 2018*) and sfe (*Curry, 2019*) for all spatial analyses. R code to repeat all the analyses is available on GitHub (https://github.com/nhcooper123/protecting-pangolins; *Cooper, Curry & Buckingham, 2020*).

To reduce the likelihood of our results being due to error in georeferencing we only included specimens with certainty scores $\geq 50\%$ and extents <50 km in our analyses. Additionally, because we are interested in understanding which localities overlap with current day ranges, rather than specimens *per se*, we excluded 107 duplicate records, i.e., where multiple specimens of one species were collected from the same locality, in all analyses except those involving collection years. For collection years analyses we instead omitted 84 duplicates where multiple specimens of one species were collected from the same locality in the same year.

### Correlates of overlaps with AOH maps

For each species of pangolin we calculated overlaps between specimen localities and present-day AOH polygons in two ways (Fig. 1). (1) Percentage locality overlap: the percentage of specimen localities (including their error extents) that fall within their species AOH polygons; (2) Percentage area overlap: the mean percentage area of specimen point-radius polygons that overlap with AOH range polygons. We calculated these percentages for specimens with certainty scores $\geq 50\%$ and extents <50 km with and without duplicates, and for all specimens with certainty scores >0%, with and without duplicates. We also grouped species by continent, ecology and IUCN Red List status and calculated the

 

percentage overlaps for these groupings. Note that *Smutsia temminckii* had an extremely complex AOH map because of its broad range, meaning that we were unable to extract overlaps from it due to computational issues (lack of available memory). We therefore used the IUCN map for this species to estimate overlaps rather than its AOH.

To test for correlates of overlaps among specimen localities and present-day range maps, we first fitted binomial generalised linear models (GLMs), with the number of specimens whose localities (including their error extents) were within their species AOH polygons (success) and the number of specimens whose localities were not within their species AOH polygons (failure) for each species as the response variable (i.e., a binomial response where the number of successes and failures were jointly modelled). We used the collection year, continent, whether the species was primarily terrestrial or arboreal/semi-arboreal, and IUCN Red List status, as predictors, and used standard model checks for GLMs (Q-Q plot, histogram of residuals, residuals vs. linear predictors, response vs. fitted values) to assess model fit. These analyses have very low power; most have only eight data points, one for each species, except the collection years analysis which has 106 data points, one for each year × species combination.

Next, we repeated these analyses using percentage area overlap for each specimen as the response variable. We used the species, collection year, continent, whether the species was primarily terrestrial or arboreal/semi-arboreal and IUCN Red List status, as predictors (note that we could not fit multiple regressions because these resulted in rank deficient matrices because species can be written as a combination of the other predictors). Because percentage area overlap is a proportion, we fitted beta regression models after first transforming percentage area overlap to remove 0s and 1s using the following equation (*Douma & Weedon, 2019*):

$$x_i^t = x_i(n-1) + 0.5/n \tag{2}$$

where $x_i^t$ is the transformed value of $x_i$, and $n$ is the total number of observations in the dataset. We then fitted our models using the betareg function in the betareg R package (*Cribari-Neto & Zeileis, 2010*) using a logit link function and with both fixed and variable precision ($\phi$). Estimators from beta regression can be biased, particularly at low sample sizes (*Douma & Weedon, 2019*; *Grün, Kosmidis & Zeileis, 2012*), so we repeated our models using the bias correction and bias reduction methods built into betareg (*Grün, Kosmidis & Zeileis, 2012*). We compared these results to our original models to ensure that bias was not influencing our conclusions.

### Anthropogenic drivers of range change or contraction

If species have changed their ranges in response to anthropogenic changes in their habitats, we expect to see negative correlations between measures of overlap in past and present-day ranges and changes in anthropogenic variables. To test this, we used beta regression as described above to determine whether percentage area overlap was correlated with changes in human population size or land-use, between 2015 and 1850, 1900, or 1950 in turn. For human population size we used changes in log population count and log population density as predictor variables. For land-use change we investigated changes in the percentage cover of forested primary land, non-forested primary land and urban land as predictor variables.
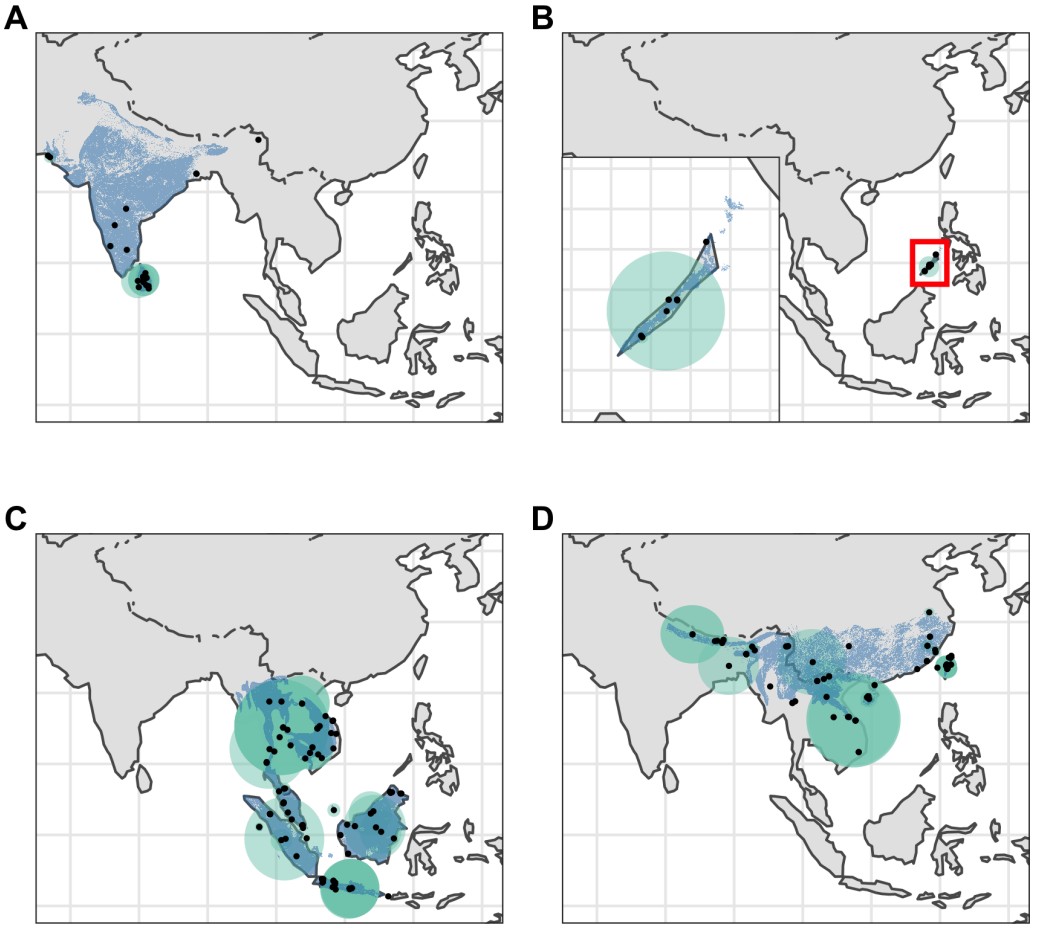

**Figure 2** **Distribution of Asian pangolins.** Point localities (black) with georeferencing error extents (green) and area of habitat maps (blue) for specimens of the four Asian pangolin species. Specimens with extents of greater than 1,000 km were omitted for clarity. (A) *Manis crassicaudata*. (B) *Manis culionensis*. (C) *Manis javanica*. (D) *Manis pentadactyla*.

## RESULTS

### Georeferencing

We georeferenced a total of 676 pangolin specimens, including 437 specimens not currently georeferenced on GBIF (Figs. 2 and 3). 269 of these had certainty scores $\geq$ 50% and extents <50 km. Of these 269, 162 represented unique collection localities and were used in our analyses (Table 1).

### Correlates of overlaps with AOH maps

Percentage locality overlaps and mean percentage area overlaps for each species, continent, ecological group and IUCN Red List status, for various subdivisions of the data, are shown in Table S3.

When modelling the number of localities overlapping with species AOH and those that did not overlap as a binomial response, we found a significant correlation with ecology

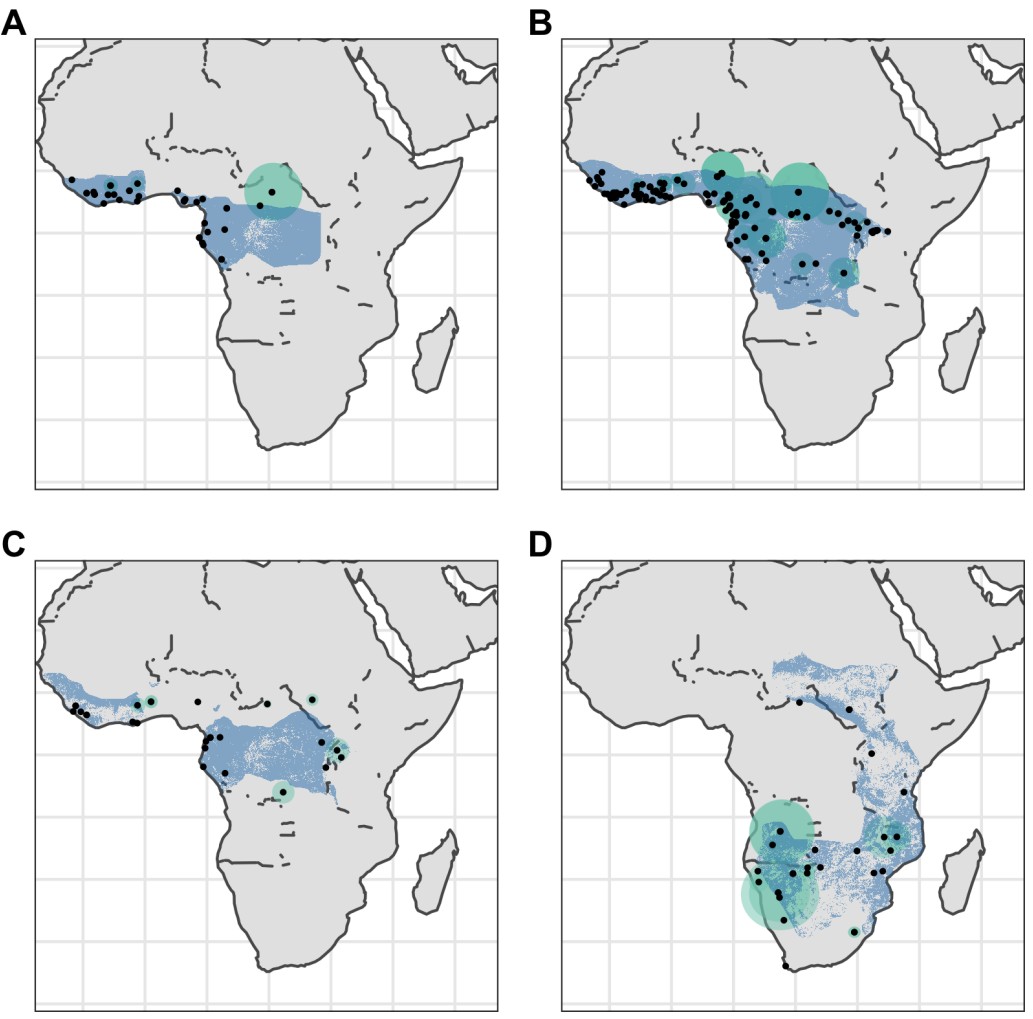

**Figure 3  Distribution of African pangolins.** Point localities (black) with georeferencing error extents (green) and area of habitat maps (blue) for specimens of the four African pangolin species. Specimens with extents of greater than 1,000 km were omitted for clarity. (A) *Phataginus tetradactyla*. (B) *Phataginus tricuspis*. (C) *Smutsia gigantea*. (D) *Smutsia temminckii*.

(binomial GLM: $\chi^2 = 4.224$, $df = 1,6$, $p = 0.040$), but not continent (binomial GLM: $\chi^2 = 2.650$, $df = 1,6$, $p = 0.104$), IUCN Red List status (binomial GLM: $\chi^2 = 0.532$, $df = 2,5$, $p = 0.766$), or collection year (binomial GLM: $\chi^2 = 1.680$, $df = 1,104$, $p = 0.195$; Fig. S1). Percentage area overlap was significantly correlated with species, continent, IUCN Red List status and collection year, but not ecology (Table S4, Fig. 4). Across species, the four *Manis* species had lower overlaps than the two *Phataginus* species, and the two *Smutsia* species (Fig. 4A). Overlaps were higher in more recent years (Fig. 4B), and higher in Africa than Asia (Fig. 4C), and in Vulnerable and Endangered species compared to Critically Endangered species (Fig. 4E).

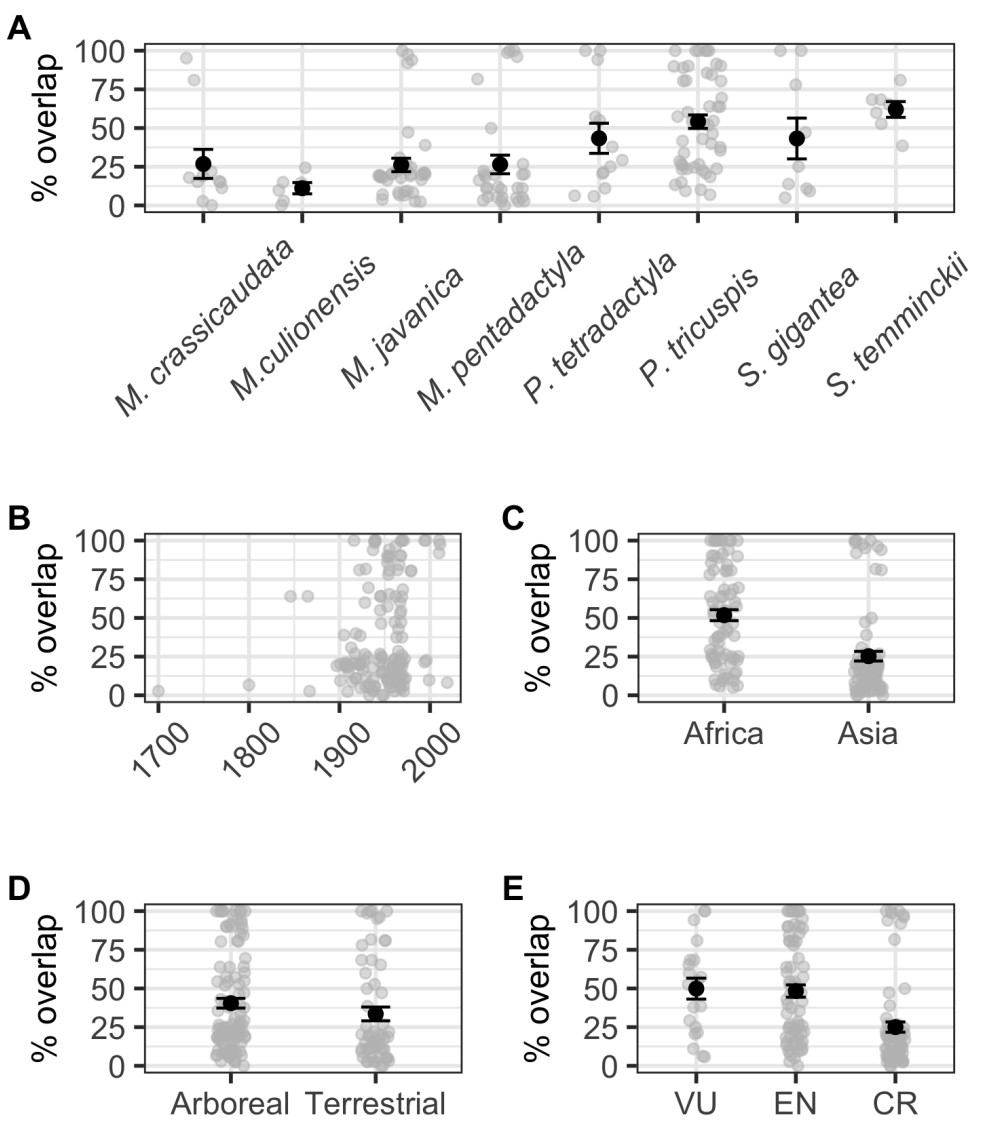

**Figure 4** **Correlates of percentage area overlaps for each specimen, for specimens with certainty scores ≥ 50%, extents < 50 km and excluding duplicates.** (A) Species. (B) Collection year. (C) Continent. (D) Ecology. (E) IUCN Red List status. Black points are means, error bars show standard errors, grey points are the raw data.

## Anthropogenic drivers of range change or contraction

There were significant negative correlations between percentage area overlaps and changes in all of the human population size variables (Fig. 5; Table S5), i.e., as population size increased, overlaps scores decreased. The pseudo-$R^2$ for these relationships, however, were very low (pseudo-$R^2$ ranged from 0.021 and 0.032) so the observed relationships between population size and degree of overlap should not be overinterpreted. Percentage area overlap was also significantly correlated with changes in the percentage cover of forested primary land between 1900 and the present, and 1950 and the present, but with no other land-use change variables (Fig. 6; Table S5). Specimens found in areas with large decreases

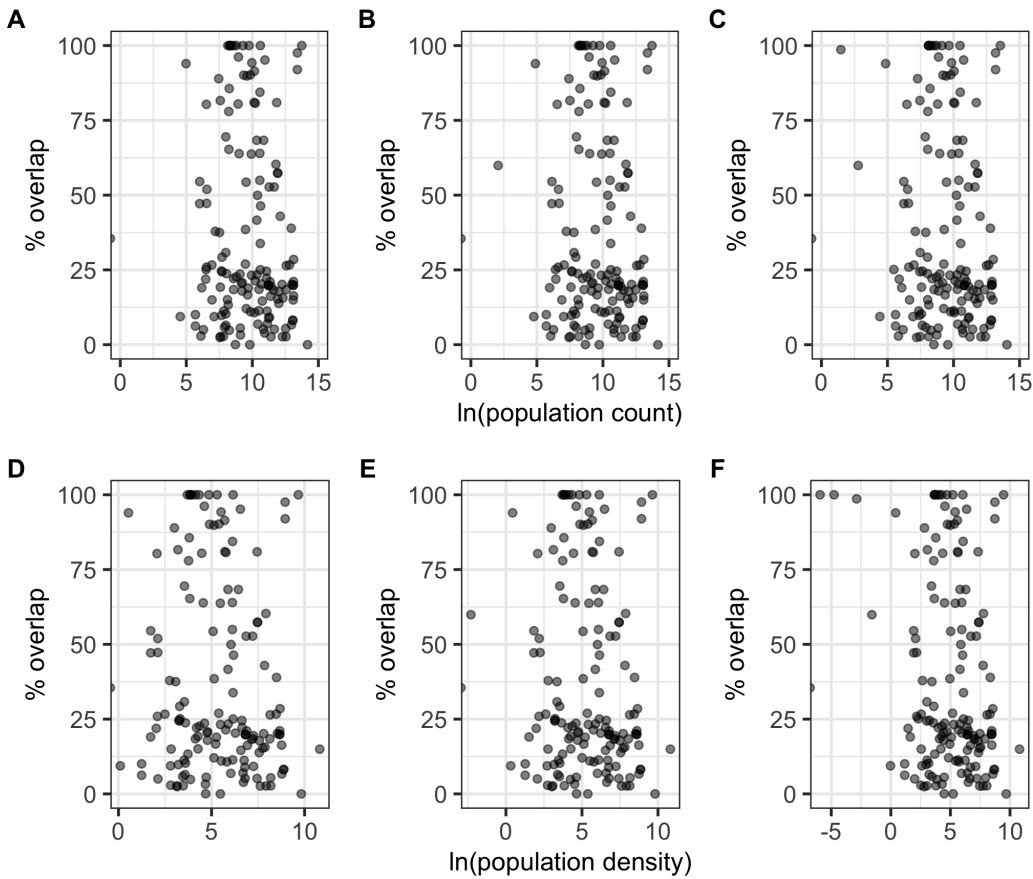

**Figure 5** **Percentage area overlap and population size.** Relationships between percentage area overlap and change in population count or population density within the point-radius polygon for each specimen between the present day and either 1850, 1900 or 1950. (A) Population count 1850. (B) Population count 1900. (C) Population count 1950. (D) Population density 1850. (E) Population density 1900. (F) Population density 1950.

in forested primary land tended to have lower overlap scores than those where forest cover had changed less. However, the pseudo-$R^2$ values for these models were also very low (0.052 and 0.033 respectively).

## DISCUSSION

Our results show that overlaps between historical localities and present-day ranges of pangolins vary across species, time, continents and extinction risk. These results are mostly driven by differences among species, with the most threatened species belonging to the genus *Manis*. Occurring in Asia, these species have the lowest percentage overlaps between their historical collection localities and present-day ranges, reflecting both suspected and estimated declines in their populations (*Irshad et al., 2015*; *Challender, Nash & Waterman, 2020*; *Wu et al., 2004*). Interestingly, species in the African genus *Smutsia* had the second lowest overlap scores, despite only being classified as Vulnerable by the IUCN Red List (*S. gigantea* is now listed as Endangered; *IUCN, 2020*). *S. gigantea* and *S. temminckii* are

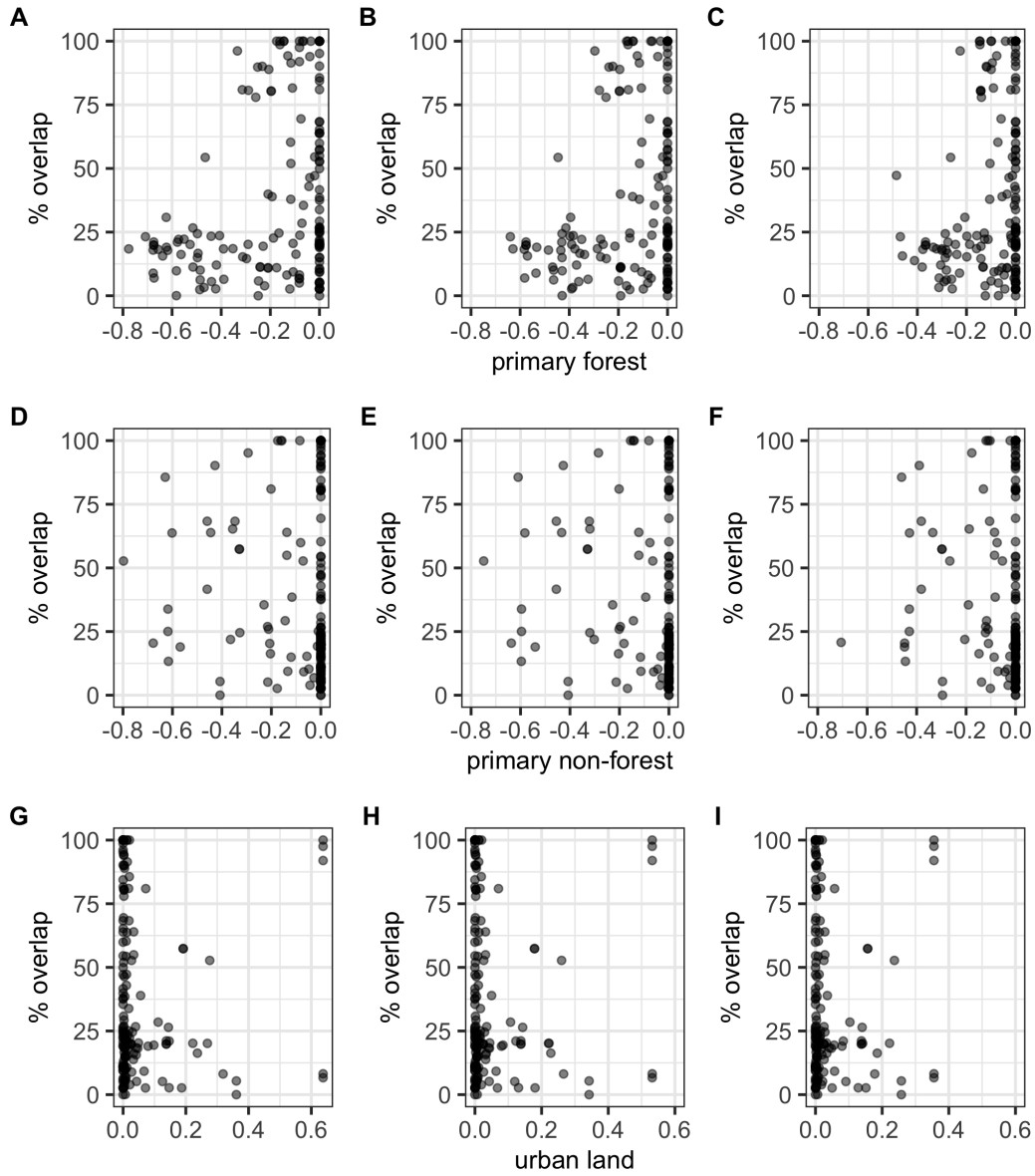

**Figure 6 Percentage area overlap and land use change.** Relationships between percentage overlap and change in primary forest cover, primary non-forest cover, and urban land cover within the point-radius polygon for each specimen between the present day and 1850, 1900 or 1950. Note that the *x*-axis is negative, showing a decrease in land cover, for primary forested and non-forested land, but positive, showing an increase in land cover, for urban land. (A) Primary forest cover 1850. (B) Primary forest cover 1900. (C) Primary forest cover 1950. (D) Primary non-forest cover 1850. (E) Primary non-forest cover 1900. (F) Primary non-forest cover 1950. (G) Urban land cover 1850. (H) Urban land cover 1900. (I) Urban land cover 1950.

large, ground dwelling and prefer relatively open landscapes so may be more accessible and attractive to poachers than smaller arboreal or semi-arboreal pangolins. In addition, the increasing pressure on African pangolin populations to meet Asian consumer demand (*Challender, Harrop & MacMillan, 2015*; *Heinrich et al., 2016*; *Shepherd et al., 2017*; *Ingram*

*et al., 2018*), which is in part due to population declines of their Asian relatives, may also be putting these larger pangolin species at a higher risk of poaching.

Correlations between overlaps and human population density—a proxy for overexploitation of species (*Woodroffe, 2000*)—had very low explanatory power. Likewise, although land-use change leading to habitat destruction further increases the vulnerability of pangolins, particularly in South East Asia which has experienced more deforestation than any other tropical region in the world (*Zhao et al., 2006*), we also did not find strong correlations between land-use change and overlaps. Instead, mechanisms of pangolin range loss are likely more complex. One root of this weak correlation appears to be that even in areas experiencing increases in human population size and urban land cover and decreases in primary forested and non-forested land cover (see Figs. 5 and 6), many specimen localities still overlap 100% with the present-day AOH range. This may be related to how we georeference locality data. Georeferencing is easier and more accurate for named places, often towns and cities, compared to more remote areas, meaning that these sorts of localities are over-represented in our data (*Buckingham et al., 2019*). This is exacerbated by the georeferencing protocol (see *Chapman & Wieczorek, 2006*) stating that localities with very small extents should have low certainty scores because of the low likelihood that the specimen was actually collected there. This meant many rural localities like farms were dropped from our analyses because their certainty scores were <50%. Likewise, many wilderness areas, for example localities in national parks, are hard to georeference accurately leading to these localities being dropped from our analyses for having extents >50 km. Taken together, this suggests we are more likely to have specimens from urban localities than from natural landscapes. Another notable problem with our data is that we can only record the presence of the species (*Tingley & Beissinger, 2009*) but neither the historical localities nor present-day range maps give any indication of species abundance or population health. A final possibility is that pangolins in many regions are able to adapt to modified habitats, for example, oil palm plantations, urban parks and gardens and rubber plantations (*Challender, Nash & Waterman, 2020*), meaning they maintain their range after human disturbance. It is unlikely, however, that such suboptimal habitat supports the same population densities as pristine forest. Thus 100% overlap for a species might actually not be such a positive thing as it first appears.

Although we assume lack of overlap among historical and present-day ranges indicates range changes or contractions, it may also be the result of error. There may be errors in georeferencing or collection notes, despite our best efforts to mitigate these. Our quality control procedures are designed to deal with uncertainty in georeferencing, but for historical data we cannot be completely sure that the correct place name was recorded (*Tingley & Beissinger, 2009*), and often the recorded name is the location of shipping or distribution, rather than the actual collection locality. Locality descriptions on specimen labels are often vague, cryptic or faded so even if the correct place was recorded, it may not be possible to recover that information. We attempted to deal with some of this in our analyses, but it is an issue that future studies should be aware of.

Error may also be present in the AOH maps which are dependent on the accuracies of the IUCN range maps, reported habitat preferences and elevational limits of each species

(*Brooks et al., 2019*). The land-use data used to create the AOH maps was created in 2018 and with rapid urbanisation and deforestation in many areas, particularly around South East Asia (*Zhao et al., 2006*), we suspect there are regions where land-use has changed since these maps were produced. Although AOH maps cannot be used to compare with either the Extent of Occurrence (EOO) or Area of Occupancy (AOO) thresholds, they provide more detail than the IUCN range maps which is useful when targeting field surveys and assessing habitat fragmentation over time (*Brooks et al., 2019*). In addition to being easy and cheap to produce with basic statistical software, a time-series of AOH maps can also be used to derive estimates of species population decline under Red List criterion A (*Buchanan et al., 2008*; *Tracewski et al., 2016*).

Museum specimens, some of which are 150 years old, are only one source of data on historical pangolin distributions. A primary challenge for pangolin conservationists is finding efficient ways to collect, collate and share data on the status of, and threats to, present-day pangolin populations. While AOH maps and other broad mapping tools (such as extent of occurrence, area of occupancy and species distribution modelling) can help overcome this challenge, it is crucial to consider other data sources such as direct or indirect field observations and sightings by community scientists. These may provide more fine-scale information on distribution patterns and provide a better understanding of how pangolin populations are changing over shorter temporal scales. However, determining how to best share these data to ensure they reach conservationists but not those seeking to exploit the species is an important concern that requires careful consideration. Other data sources, such as eDNA, and stable isotopes or molecular data from seized scales, will also be of key importance moving forwards. Physical specimens will remain critical to future conservation studies, as specimens provide far more information than a sighting, for example the age, sex, health and condition of the animals, which are all important considerations for maintaining healthy populations. Additionally, the potential applications of specimen data are continually expanding, especially with modern research techniques. Museum collections are vital for multidisciplinary research and collaborations, and have the potential to unlock previously hidden data contained within specimens using both present-day methods and currently unrealized future research techniques (*McLean et al., 2016*; *Meineke et al., 2018*).

Our study highlights the potential and limitations of using museum data to infer range changes and contractions in threatened species which can be used to evaluate present-day extinction risks when modern data is scarce. We georeferenced a total of 676 pangolin specimens, including 437 specimens not currently georeferenced on GBIF. This data is valuable to conservationists but needs to be used with care. Our results suggest that although it is not possible to accurately estimate geographic ranges from historical locality data alone, museum collections contain a wealth of spatial and temporal information that can be combined to analyse distributional trends. The most valuable data are the older records, i.e., those collected before 1900, which are not available from other sources. Future research into pangolin population trends should combine field data, data from museum collections, seizure records and local knowledge to create a full picture of declines in these species.

## ACKNOWLEDGEMENTS

We thank Jeffrey Streicher for input to the ideas that led to this project, and three anonymous reviewers for their helpful comments.

### Funding

The authors received no funding for this work.

### Competing Interests

The authors declare there are no competing interests.

### Author Contributions

- Emily Buckingham conceived and designed the experiments, performed the experiments, analyzed the data, prepared figures and/or tables, authored or reviewed drafts of the paper, and approved the final draft.
- Jake Curry and Natalie Cooper conceived and designed the experiments, analyzed the data, prepared figures and/or tables, authored or reviewed drafts of the paper, and approved the final draft.
- Charles Emogor and Louise Tomsett conceived and designed the experiments, authored or reviewed drafts of the paper, and approved the final draft.

### Data Availability

Data is available from the Natural History Museum Data Portal:

Emily Buckingham, Jake Curry, Charles Emogor, Louise Tomsett, Natalie Cooper (2019). Dataset: Pangolin georeferencing. Natural History Museum Data Portal (data.nhm.ac.uk). https://doi.org/10.5519/0058821.

Code is available from GitHub:

Cooper N, Curry J, Buckingham E. 2020. Protecting-pangolins. Github, https://github.com/nhcooper123/protecting-pangolins. Code for the paper. Zenodo. http://doi.org/10.5281/zenodo.4266580.

R package sfe (Version v1.0) is available from Zenodo:

Jake Curry (2019). sfe (Version v1.0). Zenodo. http://doi.org/10.5281/zenodo.3578490.

### Supplemental Information

Supplemental information for this article can be found online at http://dx.doi.org/10.7717/peerj.10843#supplemental-information.

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
