# Peer review of "Using natural history collections to investigate changes in pangolin (Pholidota: Manidae) geographic ranges through time"

_PeerJ, doi:10.7717/peerj.10843_

## Round 0.1 · original submission · Major Revisions

Thank you for submitting your manuscript. I received comments from three reviewers on your manuscript.

All reviewers agreed that your paper is relevant and has good potential. Nevertheless, they suggested several points for improvement, including a major restructuring (all reviewers), additional analyses (reviewer 2), and a reader-friendly approach (all reviewers). I believe that you have excellent feedback to improve your manuscript significantly.

When resubmitting your manuscript, please carefully consider all issues mentioned in the reviewers' comments, outline every change made point by point, and provide suitable rebuttals for any remarks not addressed. Your paper will be sent again to the same reviewers.

Reviewer 1 ·

Basic reporting

A generally well presented MS, but a number of specific points ought to be addressed subject to the trajectory of this MS.

Specific points:
L42 – most species are nocturnal. Exception is the P. tetradactyla. This should be caveated.
L43-45 - There is debate on whether pangolins are the most trafficked mammals. Consider revising to ‘are considered to be…’ You may need to amend references for this point.
L44-49 – I think you are referring to international trade reported to CITES in this sentence. It cannot be said based on available evidence and the literature that most ‘trade’ has involved the Asian species. African species – like the Asian pangolins – have been traded and consumed at a local level through history. You could amend this to international trade reported to CITES and cite Challender et al. 2015 and/or Heinrich et al. 2016.
L46 – I would state large populations collapses in parts of Asia. Available evidence indicates the Chinese, Sunda and Philippine pangolin have been exploited at a higher rate than the Indian pangolin.
L46 – Need to change to apparent increases in exploitation of African pangolins if referring to international trade – legal or illegal – because while it appears this is the case, we don’t have unbiased data indicating that this is so. If you want to cite increases in exploitation referring to local use as well in Africa cite Ingram et al. 2018.
L47 – I would remove reference to Aisher. It is redundant here.
L50-51 – there are issues with captive breeding, but they don’t prevent such programmes from exiting or breeding pangolins. Taipei zoo breeds Chinese pangolins for example. Suggest revising this wording.
L52-54 – The Red List assessments are based on time periods looking forwards and backwards. Please check whether this paragraph reflects this or needs amending to reflect this. I suspect the latter.
L56. Illegal trafficking as opposed to legal trafficking. This requires editing.
L57- what does regional mean in this context? Within a country, or countries collectively (e.g., W Africa).
L59 – what specifically do you mean by stricter regulations in recent years? Do you mean amended laws? Movement of pangolins to CITES App I?
L60 – I would remove reference to Chin and Pantel 2009. This is now eleven years old and not the best source for this. See this document which discusses point explicitly: https://cites.org/sites/default/files/eng/com/sc/69/E-SC69-57-A.pdf
L61 – the actual extent of illegal trade is unknown. This is not unique to pangolins of course but applies to virtually all trafficked wildlife derivatives and other commodities. I disagree that species distributions are unknown. They are being honed cumulatively based on new, local data. Updated distributions were published in the IUCN RL in 2019. This requires editing.
L90 – typo
L105-106 – I’m not sure Mittermeier and Wilson (2011) is the best source for this information. New authoritative sources have been published on pangolins which contradict this table for white-bellied pangolin. See https://www.sciencedirect.com/science/article/pii/B9780128155073000095.
L138 – what resolution?
L154- population size of land-use variable, but not both, correct?
L165-169 – MS reverts to alternative font.
L170 – how many specimens/record did this rule result in being omitted?
L233-234 – Makes me wonder why all these records are not in GBIF!
L256 – why highlight this specimen specifically here?
L256-259 – Anything else to say about these results? It reads like you have given them short shrift.
L263 – extinction risk. No need for ‘status’ here. Or, say IUCN RL status.
L264 – occur in?
L271-273 – This oversimplifies things a little. See previous comments on trade characterisation. There appears to be increasing pressure on African populations. See Ingram et al. 2018. Aisher et al. 2016 is again redundant here.
L275 - There are two species of Smutsia - are you referring to them both here?
L278-280 – I think you are referring to Asia here, but I’m guessing because it is not clear. In Asia the meat is considered a delicacy by some in countries including Vietnam and China, but is also consumed locally around protected areas too. Again Aisher et al. 2016 doesn’t add any value here.
L309-311 - This is possible – though it is worth noting here that the maps were updated in 2019, which you will need to recognise here.
L323-330 – Although these measures (and the omission of certain data points) were designed to deal with error as far as possible.
L333 – I think you mean threatened species here, no just endangered (=Endangered) here?
L340-342 – I’m not sure this is a fitting final sentence for reasons similar to the above point. Suggest reviewing.
Fig. 2 – I suggest zooming into the Palawan faunal region for the Philippine pangolin. Currently, this map is not particularly useful but it is hard to see where the points are in the species range.
Fig. 3 – New RL maps available from the RL version 2019-3.

Experimental design

I have some specific thoughts/questions on experiment design:
L74-77 – why weren’t other museum records included in the analyses? Why was this limited to GBIF+NHM London?
L103-104 – Why did you use range maps from 2014, and RL status from 2020? Why not use the range maps published with the new Red List assessments in December 2019? This would make the analyses as up to date as possible.
L103-104 – The map for Manis crassicaudata is incorrect and should not reflect distribution in Myanmar. This is an error on the IUCN Red List account for that species. It is not an insignificant error in my opinion and ought to be corrected.
L148 – I suggest the present day be 2019 using updated RL maps. As this paper – if accepted – would be published in 2019, using 2014 (i.e., old maps) need justification.
L284-293. I think this slightly mischaracterises habitat use and occupancy of pangolins in modified landscapes. While habitat loss does pose a threat, e.g., in SE Asia, the Sunda pangolin, like the white-bellied pangolin (and others) is able to adapt to modified habitats e.g., oil palm plantations, gardens and rubber plantations etc. This is documented in the literature (see here https://www.sciencedirect.com/science/article/pii/B978012815507300006X?via%3Dihub and the RL assessment for this species). However, there is little detailed knowledge on things like relative densities between tropical forest and oil palm plantations. I make this point because one could have perhaps been able to predict – based in part on this – that this would have been the outcome of this research. I suggest this and similar insights be captured in the discussion in Lines 290-319.
L334 – Fine, but most of these 437 were not used in the analyses, correct?

Validity of the findings

Some specific points:
L277-278 – Doesn’t this seriously undermine the framework being proposed in the paper? To be honest, I’d be surprised if you had found a mechanism using these data.
L337 – I think this would have been a remarkable finding if it were. In fact, I’m not sure that hypothesis that it would have been would have been a sound one.
L338 – 340 – This feels like a limp ending. It reads as though, despite your efforts, you are saying that the real value in using museum data is to use the old records because they are not available anywhere else. The MS needs another couple of sentences that really underlines the potential of this research to inform contemporary conservation.
Overall - I think the paper is flawed because it does not place the findings of the modelling work in a realistic context for pangolins, i.e., relating the findings adequately to the species ecology. The study would be have been much stronger, I suspect, had the authors collaborated with researchers that understand pangolins and their ecology.
Fig. 2 – The range map for the Indian pangolin is incorrect. I recommend downloading the corrected version from the IUCN RL 2019-3 and using that (alongside new maps for all species).

Additional comments

This is an interesting study but I am not convinced in its current form the MS meets the requirements for publication. The MS proposes use of framework but doesn't really present a framework. It uses a novel method but seems to conclude that the best use of museum data is as older locality records. Nor does it place the findings of the modelling work in a realistic context for pangolins.

Reviewer 2 ·

Basic reporting

The paper is well written, the analyses carefully done and clearly explained, and the figures and tables clear and well documented. There are some minor errors in the literature cited.The data are also available for reanalysis.

Experimental design

I think that some additional analyses would make the paper much more sound scientifically. It is not a question of an incorrect approach as stands, just that there is an opportunity to enhance the results.

Validity of the findings

Nothing really to add here, but I will add my comments and suggestions relevant to this point below.

Additional comments

I have made a number of comments on the pdf attached. These tend to be more detailed comments and suggestions. I have two additional broader suggestion.

The Introduction seems to end with a presentation of results, even some discussion. This does not fit well, and it would be better to tie the introduction into the literature on evaluation of threats, threat assessments, and the Red List process. This links the paper more clearly to the process of improving threat assessments, especially in a spatial context. The ending of the introduction needs to be revised to address this.There are a number of more recent papers the discuss the improvement of threat evaluation (e.g. some of the work of Lucas Joppa).

In the section lines 300-320, there is the statement "solution would be to use IUCN range maps along with information on habitat suitability" however a group from IUCN recently published a paper with explicit instructions on how to do this, developing a "Area of Habitat" map. here is the citation.

Brooks, T.M., Pimm, S.L., Akçakaya, H.R., Buchanan, G.M., Butchart, S.H.M., Foden, W., Hilton-Taylor, C., Hoffmann, M., Jenkins, C.N., Joppa, L., Li, B.V., Menon, V., Ocampo-Peñuela, N., Rondinini, C., 2019. Measuring Terrestrial Area of Habitat (AOH) and Its Utility for the IUCN Red List. Trends in Ecology & Evolution 34, 977–986. https://doi.org/10.1016/j.tree.2019.06.009

I think that the quality of the analysis, and its relevance to Red List assessments, would be greatly enhanced by calculating AOH from the Red List maps used in the paper, and rerunning the analyses. I have done AOH recently in a paper submitted, and it is not complicated. The paper as it stands is fine, but with a lot of caveats. I think this would greatly improve the conservation values of this analysis but updated the maps to deal with the authors expressed concerns about problems with the IUCN polygons. AOH is far less complicated than running SDMs, but with a similar impact in improving the results.

Annotated reviews are not available for download in order to protect the identity of reviewers who chose to remain anonymous.

Reviewer 3 ·

Basic reporting

This paper is well written but needs more description of the taxa in the text for readers to be drawn into the project. Some of the analyses could be described with a few additional details to help readers not familiar with how this science is done.

Experimental design

No comment

Validity of the findings

This is a timely topic to be looking into. The authors have clearly done an extensive amount of work georeferencing the specimens in the NHM collections which is valuable. The way they assess the accuracy of historical records is appropriate. Effort was put into incorporating different types of data to add to their comparisons about the analyses and testing hypotheses with the available data (which unfortunately are few).
Having read through this, however, I feel that the authors may have missed the most important aspect of what their data show and why. The specific question being asked was whether or not there was a change in presence/absence over time (and the answer is "not that they could detect change"), but the true problem is not with the historical data, it is with the modern data (Note, that I feel that this was at least an indirect message given by Boakes et al. in their work). We basically do not have suitable modern collections (or data points on presence/absence) for pangolins to say much of anything about changes in distribution in the present during the current intense harvesting pressure which they note. This is, of course, stunning given how many have been caught for trade. As I read the paper, the authors are assuming "present day" data in the form of IUCN range maps are somehow updated and "modern" and I highly doubt they are. I didn't see anywhere in this manuscript where the way that IUCN ranges were determined (and the broad geographic scale is an important factor) is described to any extent. I am guessing that experts were involved, but I am also guessing that a good portion of the same historical records was incorporated (or at least basic knowledge of those data) into generating the IUCN range maps.
I have been criticized by reviewers for using shape files of ranges in my own research and I believe they actually reflect well most of the time, ranges, but that is because well-constructed polygons of ranges include historical data, which makes them useful for some research questions and not for others. I think these present data and their analyses can be published (and should be), but the message should be how critical it is for there to be surveys conducted regionally whereby data is gathered about the modern status of these species within these polygons. Personally, I believe this should include getting modern specimens to, in part large part because I think it is likely there is substantial population structure within many of these species. These populations may be differentially affected. Thus, although the historic specimens tell you about historical patterns, only with similar modern-day data can you truly assess change over time (and this has to be investigated across different scales. For instance, my guess is that the range of and of these species on an island in SE Asia is smaller because of habitat loss, but I don't this study was designed at all to capture that. What the authors should advocate for is the need for modern data (and specimens), because I think there will be ample evidence for areas where they no longer occur (and that those areas will related to human activities.

The scale at which this study was done is not overly interesting because the current range maps are likely based on roughly the same historical data.

Line 71. delete "can," these records to provide data from before exploitation was so widespread.
Line 76. What are present day range maps based on?
Line 97. Give specific download information that GBIF provides.
Line 137. Grid cell is brought up for the first time here, give more information on the term. Grid cell size would seem to me to have potentially huge influence of interpretations of pattern.
Line 171. Do you think anything more could be done with data on multiple specimens in terms historical abundances?
Line 196. Would using decade to bin records have increased power?
How were the IUCN range maps made?

Help readers out. Introduce pangolins by taxonomy, natural history, and current threat status at the beginning of the manuscript.
Line 256. I am very interested in these kinds of projects, but I have know idea what the significance of this sample is. Help readers out here. How could there possibly not have been those data for this locality?

Line 274. Fix this sentence to clarify whether you are referring to the genus in general or one species in particular here.

Line 314. The statement about IUCN range maps illustrates a huge challenge for this paper. It seems highly likely to me that these maps were made using relying extensively on the historical records, so there is complete circularity here.

Additional comments

This paper would best be restructured to highlight that historical records reflect percieved present distributions of these taxa well, but we are in great need of better modern data.

---

## Round 0.2 · Minor Revisions

Both reviewers think your paper has improved and can be accepted for publication. However, reviewer 3 suggests that you should highlight what kinds of data are needed going forward. I agree with the reviewer. Please, add your ideas to the paper and submit an updated version as soon as possible.

Reviewer 2 ·

Basic reporting

This is the revision of a previous version. I have reviewed them both and this version is significantly improved. It is clearly written, well cited using the appropriate literature, uses solid statistical and spatial analytical approaches with maximal reduction in potential biases, and provides access to data and code. The figures are also clear and easy to interpret.

Experimental design

Well conceived and carefully explained in the Methods. This is much improved over the previous version of the manuscript. It asks important questions about understanding factors responsible for the decline in pangolin populations and how we might better track these using historical records.

Validity of the findings

The findings are supported by the data and analyses and the authors present the appropriate cautions in interpreting the results, given possible biases and errors in the raw data (not the responsibility of the authors).The paper does highlight the important value of museum specimen records. This has the additional importance of presenting the role of natural history museums when so many are at threat.

Additional comments

I am quite please with the responses to all of the prior comments and suggestions. The inclusion of AOH has greatly improved the manuscript.

Reviewer 3 ·

Basic reporting

I commend the authors for responding to the comments with changes that have greatly improved the manuscript.

Experimental design

This is fine

Validity of the findings

This is fine

Additional comments

I continue to be concerned that they have not really used their results to address the primary concern going forward for documenting pangolin distribution change and that is gathering more than what seems to be quite limited data on the presence/absence of these species across their distributions. The AOH maps are still based on limited knowledge (that is not really quantified in any documented way). This an opportunity to discuss that (even if the crux of the paper is to use the historical records). One potential way forward would essentially be e-mammal, which would be the creation of a citizen science database along the lines of e-bird. This doesn't obviate the need for specimens because specimens harbor data that go far beyond a sighting (something not really brought up anywhere in the manuscript), but if one is simply trying to assess potential changes in distribution through time, these approaches will gather data that can be applied across the broad geographic (and subsequently temporal) scales being considered here to answer important questions about how pangolins are responding to landscape change through time which is a start.

---

## Round 0.3 · accepted · Accept

Congratulations! Please work with our production team to get your paper published.